

# Evaluation of OH and HO$_2$ concentrations and their budgets during photo-oxidation of 2-methyl-3-butene-2-ol (MBO) in the atmospheric simulation chamber SAPHIR

Anna Novelli[1], Martin Kaminski[1,a], Michael Rolletter[1], Ismail-Hakki Acir[1,b], Birger Bohn[1], Hans-Peter
Dorn[1], Xin Li[1,c], Anna Lutz[2], Sascha Nehr[1,d], Franz Rohrer[1], Ralf Tillmann[1], Robert Wegener[1], Frank
Holland[1], Andreas Hofzumahaus[1], Astrid Kiendler-Scharr[1], AndreasWahner[1] and Hendrik Fuchs[1]

[1]Institute of Energy and Climate Research, IEK-8: Troposphere, Forschungszentrum Jülich GmbH, Jülich, Germany
[2]Department of Chemistry and Molecular Biology, University of Gothenburg, Gothenburg, Sweden
[a]now at: Bundesamt für Verbraucherschutz, Abteilung 5 – Methodenstandardisierung,
Referenzlaboratorien und Antibiotikaresistenz, Berlin, Germany
[b]now at: Institute of Nutrition and Food Sciences, Food Chemistry, University of Bonn, Bonn, Germany
[c]now at: State Key Joint Laboratory of Environmental Simulation and Pollution Control, College of Environmental Sciences
and Engineering, Peking University, Beijing, China
[d]now at: INBUREX Consulting GmbH, Process Safety, Hamm, Germany

*Correspondence to*: Anna Novelli (a.novelli@fz-juelich.de) or Hendrik Fuchs (h.fuchs@fz-juelich.de)

**Abstract.** Several field studies reported unexpected large concentrations of hydroxyl and hydroperoxyl radicals (OH and HO$_2$, respectively) in forested environments that could not be explained by the traditional oxidation mechanisms which largely underestimated the observations. These environments were characterized by large concentrations of biogenic volatile organic compounds (BVOC) and low nitrogen oxide concentration. In isoprene-dominated environments, models developed to simulate atmospheric photochemistry generally underestimated the observed OH radical concentrations. In contrast, HO$_2$ radical concentration showed large discrepancies with model simulations mainly in non-isoprene dominated forested environments. An abundant BVOC emitted by lodgepole and ponderosa pines is 2-methyl-3-butene-2-ol (MBO), observed in large concentrations for studies where the HO$_2$ concentration was poorly described by model simulations. In this work, the photooxidation of MBO by OH was investigated for NO concentrations lower than 200 pptv in the atmospheric simulation chamber SAPHIR at Forschungszentrum Jülich. Measurements of OH and HO$_2$ radicals, OH reactivity ($k_{OH}$), MBO, OH precursors and organic products (acetone and formaldehyde) were used to test our current understanding of the OH-oxidation mechanisms for MBO by comparing measurements with model calculations. All the measured trace gases agree well with the model results (within 15%) indicating a well understood mechanism for the MBO oxidation by OH. Therefore, the oxidation of MBO cannot contribute to reconcile the unexplained high OH and HO$_2$ radical concentrations found in previous field studies.

## 1 Introduction

The hydroxyl radical (OH) is the most important daytime oxidant in the troposphere and its concentration affects the fate of many pollutants thus having a direct impact on the formation of ozone (O$_3$), oxygenated volatile organic compounds (OVOCs) and as such, influencing particle formation and climate. Understanding the OH radical formation and destruction paths is therefore critical.

Measurements of OH radicals in environments characterized by low NO concentrations, pristine conditions and isoprene being the most abundant measured BVOC (Carslaw et al., 2001; Tan et al., 2001; Lelieveld et al., 2008; Whalley et al., 2011; Wolfe et al., 2011a), as well as in environments characterized by mixed emissions from biogenic and anthropogenic sources (Hofzumahaus et al., 2009; Lu et al., 2012; Lu et al., 2013; Tan et al., 2017), have shown a significant underestimation of observed OH concentrations by state-of-the-art models. In addition, the analysis of the OH budget using only measured



species obtained by comparing all known OH radical sources together with the OH radical loss rate has demonstrated that the discrepancy with model simulations is due to a large missing OH radical source (Rohrer et al., 2014). Theoretical studies have proposed new OH recycling paths which, contrary to traditional mechanisms, do not require NO for the regeneration of $HO_x$ from $RO_2$ radicals. The proposed mechanism involves unimolecular reactions of specific isoprene peroxy radicals

($RO_2$) (Dibble, 2004; Peeters et al., 2009; Nguyen et al., 2010; Peeters and Müller, 2010; Silva et al., 2010; Peeters and Nguyen, 2012; Peeters et al., 2014). Laboratory (Crounse et al., 2011) and chamber studies (Fuchs et al., 2013) have confirmed this mechanism and have helped constraining its atmospheric impact. Contemporary, other trace gases have been investigated as the results from the isoprene studies show that OH recycling through isoprene-$RO_2$, alone, is not sufficient to explain the OH concentrations observed in the field. Chamber and laboratory studies on methacrolein (MACR) (Crounse et

al., 2012; Fuchs et al., 2014), methyl vinyl ketone (MVK) (Praske et al., 2015), isoprene hydroxy hydroperoxide (D'Ambro et al., 2017) and glyoxal (Feierabend et al., 2008; Lockhart et al., 2013), important products from the oxidation of isoprene by OH, also have shown new OH recycling paths as predicted by theory (Peeters et al., 2009; da Silva, 2010; Setokuchi, 2011; da Silva, 2012). Further laboratory studies also have discovered OH radical recycling in the bimolecular reaction of $HO_2$ with acyl peroxy radicals which was previously considered to be a radical termination reaction only (Dillon and

Crowley, 2008; Groß et al., 2014; Praske et al., 2015). These results underline the need to carefully investigate the OH radical budget, at least for the most abundant volatile organic compounds, to test our current knowledge.

In a similar way, the $HO_2$ radical concentration measured in several field campaigns performed in forested areas have shown discrepancy of measurements with model calculations highlighting an incomplete understanding of the chemistry involving formation and loss paths of $HO_2$ radicals. In some environments, the model tends to overestimate the measured $HO_2$

concentration (Stone et al., 2011) while in others there is the tendency to underestimate the measurements (Kubistin et al., 2010; Wolfe et al., 2011b; Hens et al., 2014; Wolfe et al., 2014). It has been recently shown that $HO_2$ radical measurements performed by laser induced fluorescence (LIF) via conversion of $HO_2$ into OH after reaction with nitrogen oxide (NO) are likely affected by an interference that originates from organic peroxy radicals (Fuchs et al., 2011; Nehr et al., 2011; Whalley et al., 2013; Lew et al., 2018). Therefore some of the discrepancies observed in previous studies may be partly caused by

inaccurate $HO_2$ radical measurements. Nevertheless, recent studies where this peroxy radical interference is accounted for still showed discrepancy with model calculations (Griffith et al., 2013; Hens et al., 2014; Wolfe et al., 2014) These recent studies, performed in environments where isoprene was not the dominant measured BVOC, were all characterized by poor agreement between modelled and measured results for both OH and $HO_2$ concentrations, with the measurements being up to a factor of 3 higher than the model results. Good agreement was observed between modelled and measured OH radicals

when the model is constrained to the $HO_2$ radical measurements. These studies have concluded that there is a missing $HO_2$ source for environments where the dominant measured BVOC are monoterpenes and 2-methyl-3-butene-2-ol (MBO) (Hens et al., 2014; Wolfe et al., 2014). Corresponding photo-oxidation studies have been performed for β-pinene in the SAPHIR chamber at Forschungszentrum Jülich. Consistent with field studies, a significant (up to a factor of 2) model under-prediction of both OH and $HO_2$ concentrations was observed when β-pinene was oxidized by OH under low-NO conditions

(< 300 pptv) (Kaminski et al., 2017). The observed discrepancies in the chamber could be explained by additional $HO_2$ production, for which Kaminski et al. (2017) proposed a mechanism involving unimolecular radical reactions and photolysis of oxygenated products.

MBO is the dominant emission from lodgepole (*Pinus Contorta*) and ponderosa (*Pinus Ponderosa*) pines (Goldan et al., 1993; Harley et al., 1998). Its global emission is lower than that of isoprene (Guenther et al., 2012), but in forested areas

within the West-US, MBO is the most abundant BVOC measured contributing most of the measured OH reactivity (Nakashima et al., 2014; Ortega et al., 2014). In the atmosphere, MBO reacts primarily with OH forming two peroxy radicals that yield acetone, glycolaldehyde, 2-hydroxy-2-methylpropanal (HMPR) and formaldehyde after reaction with NO (Fantechi et al., 1998; Ferronato et al., 1998; Carrasco et al., 2006) (Fig. 1). The reaction of the peroxy radicals with $HO_2$





yields two different dihydroxy hydroperoxides (MBOAOOH and MBOBOOH, Fig. 1). Recent theoretical studies have described a mechanism that involves additional hydrogen shift reactions for the $RO_2$ that reforms OH and produces $HO_2$ (Knap et al., 2015; Knap et al., 2016). As the predicted unimolecular reaction rate following the hydrogen shift is, at most, 1.1 x $10^{-3}$ $s^{-1}$ (at 298 K and 1013 hPa) the study by Knap et al. (2016) concludes that in environments where the NO

concentration is high (> 1 ppbv), the reaction between $RO_2$ and NO will be the dominant loss path for $RO_2$ radicals, and in forested areas, where the NO concentration is lower than 0.2 ppbv, reactions with $HO_2$ and $RO_2$ will dominate the $RO_2$ fate. In this study, the photo-oxidation of MBO initiated by OH radicals is investigated in the atmospheric simulation chamber SAPHIR in the presence of approximately 200 pptv of NO. The OH and $HO_2$ budget are analysed and a comparison with an up-to-date model is performed to test the current understanding of the oxidation processes of this important BVOC.

## 2 Methods

### 1.1 Atmospheric simulation chamber SAPHIR

The experiment performed in this study was conducted in the atmospheric simulation chamber SAPHIR at Forschungszentrum Jülich, Germany. The chamber allows for the investigation of oxidation processes and mechanisms of

organic compounds at atmospheric conditions in a controlled environment. The SAPHIR chamber has a cylindrical shape with a volume of 270 $m^3$ and is made of a double-wall Teflon (FEP) film that is inert and has a high transmittance for solar radiation (Bohn and Zilken, 2005). It is equipped with a shutter system that is opened during photolysis experiments allowing the natural solar radiation to penetrate the chamber. The air provided to the chamber is mixed from ultra-pure nitrogen and oxygen (Linde, > 99.99990 %). A fan in the chamber ensures a complete mixing of trace gases within two

minutes. The pressure in the chamber is slightly higher than ambient pressure (~ 30 Pa) to avoid external air penetrating the chamber. Due to small leakages and air consumption by instruments, a dilution rate of ~ 4 % $h^{-1}$ was required during this study. More details regarding the chamber can be found elsewhere (Rohrer et al., 2005; Poppe et al., 2007; Schlosser et al., 2007).

### 1.2  Experimental procedure

At the beginning of the experiment, only synthetic air was present after flushing during the night. Evaporated Milli-Q® water was first introduced into the dark chamber by a carrier flow of synthetic air containing until a concentration of ~ 5 x $10^{17}$ $cm^{-3}$ of water vapour was reached. Ozone produced by a silent discharge ozonizer (O3onia) was subsequently added to reach 40 pbbv in the chamber. This initial phase is defined as the dark phase. After opening the shutter system of the chamber, nitrous acid (HONO) was photochemically formed on the Teflon surface and released into the chamber. Its subsequent photolysis

produced OH radicals and NO (Rohrer et al., 2005) during this so-called zero-air phase. Ozone (~ 40 ppbv) was injected to keep the concentration of NO within a few hundreds of pptv. Afterwards, the MBO was injected three times at intervals of about two hours using a high-concentration gas mixture of MBO (~ 600 ppm, 98% from Merck) pre-mixed in a Silcosteel Canister (Restek) to reach ~ 4 ppbv of MBO in the chamber after each injection.

### 1.3  Instrumentation

Table 1 summarizes the instruments available during the experiment quoting time resolution, 1σ accuracy, and precision for each instrument. The concentrations of OH and $HO_2$ radicals were measured with the laser induced fluorescence (LIF) instrument permanently in use at the SAPHIR chamber and described previously (Holland et al., 2003; Fuchs et al., 2011). Recent studies have shown the possibility of interferences on the OH signal measured by LIF instruments that depend both on the chemical conditions of the sampled air and on the geometry of the different instruments (Mao et al., 2012; Novelli et



al., 2014; Rickly and Stevens, 2018). A laboratory study performed with this LIF instrument (Fuchs et al., 2016) showed only interferences for high ozone concentrations (300 ppbv – 900 ppbv) together with BVOC concentrations ranging 1 to 450 ppbv which are far beyond any condition encountered in this study. Therefore, the OH radical concentration measured by the LIF instrument in this study is considered free from interferences. In addition, OH was measured by differential

optical absorption spectroscopy (DOAS) (Dorn et al., 1995). Numerous intercomparisons between the LIF and the DOAS instrument in the SAPHIR chamber (Schlosser et al., 2007; Schlosser et al., 2009; Fuchs et al., 2012) showed very good agreement amid these two instruments giving high reliability to the OH radical measurements performed in the chamber. OH concentration measurements by DOAS in this study were on average 14% lower than those by LIF. This difference was well within the combined accuracies of measurements and was taken into account as additional uncertainty of OH concentration

measurements in the analysis of this study, for which mainly OH data from the LIF instrument was used.

Several studies have proven that $RO_2$ radicals can cause an interference signal in the $HO_2$ radicals measured by conversion to OH after reaction with an excess of NO (Fuchs et al., 2011; Hornbrook et al., 2011; Whalley et al., 2013; Lew et al., 2017). It was shown that a reasonable approach for avoiding the interference is to lower the concentration of NO reacting with the sampled air inside the instrument. During this study, the NO concentration ($\sim 2.5 \times 10^{13}$ cm$^{-3}$) was thus adjusted to lower the

interference to a minimum as described in Fuchs et al. (2011). As during the investigation of the interference from $RO_2$ originating from the oxidation of different VOCs, MBO was not tested, the amount of interference that arises from its oxidation products is not known (Fuchs et al., 2011; Whalley et al., 2013; Lew et al., 2017). An upper limit could be estimated from experiments with isoprene peroxy radicals at similar operational conditions of the instrument. The relative detection sensitivity for $RO_2$ radicals originated from isoprene, compared to the $HO_2$ signal, was among the largest of the

studied peroxy radical species, with a value of 20% under the conditions of the present work (Fuchs et al., 2011). This value is considered to be a reasonable estimate bias in the $HO_2$ radical measurements and will be considered later in the discussion. The OH reactivity ($k_{OH}$), the inverse lifetime of OH, was measured by a pump and probe technique coupled with a time-resolved detection of OH by LIF (Lou et al., 2010; Fuchs et al., 2017). MBO was measured by a proton-transfer-reaction time-of-flight mass spectrometer (PTR-TOF-MS, (Lindinger et al., 1998; Jordan et al., 2009)) and a gas chromatography

system (GC, (Wegener et al., 2007)) with a PTR-TOF-MS to GC ratio of 1.1 ± 0.1, and acetone by PTR-TOF-MS. Formaldehyde (HCHO) was detected with a Hantzsch monitor (Kelly and Fortune, 1994), HONO with a long-path absorption photometry (LOPAP (Li et al., 2014)), carbon monoxide (CO) with a reduction gas analysis instrument (RGA, (Wegener et al., 2007)), carbon dioxide ($CO_2$), methane ($CH_4$), water vapor by an instrument applying cavity ring-down spectroscopy (CRDS, Picarro), NO, nitrogen dioxide ($NO_2$) with chemiluminescence (CL, (Ridley et al., 1992)) and $O_3$ by

UV absorption (Ansyco). Photolysis frequencies were calculated from measurements of solar actinic radiation by a spectroradiometer (Bohn et al., 2005; Bohn and Zilken, 2005).

### 1.4    Model calculations

A zero-dimensional box model using chemical mechanistic information from the Master Chemical Mechanism, MCM

version 3.3.1 (Jenkin et al., 1997; Saunders et al., 2003) downloaded via website: http://mcm.leeds.ac.uk/MCM was used to calculate radical and trace gas concentrations. The model was implemented with specific chamber-related properties. First, a dilution rate was applied to all the trace gases present in the model to account for the dilution from the replenishing flow. The background production of HONO, HCHO and acetone known to occur in the sunlit chamber (Rohrer et al., 2005; Karl et al., 2006), was parametrized by an empirical function that depends on temperature, relative humidity and solar radiation.

Source strengths were adjusted to match the time series of HCHO and acetone during the zero-air phase, when the chamber was the only source for these species ($\sim 0.3$ ppbv h$^{-1}$). These chamber sources also impacted the OH reactivity measured during the zero-air phase. Ideally, after accurately accounting for the chamber sources, the OH reactivity should be well





represented by the model, but it is commonly the case that there is still the need for an OH reactant equivalent to ~ 1.0 s$^{-1}$ of OH reactivity. This unexplained reactivity is parametrized with a co-reactant Y added to the model, which converts OH to HO$_2$ in the same way as CO does (Fuchs et al., 2012; Fuchs et al., 2014; Kaminski et al., 2017). The concentration of Y was adjusted to match the observed OH reactivity during the zero-air phase of the experiment and was kept constant throughout

the experiment. The uncertainty of the OH reactivity in this experiment was ± 0.6 s$^{-1}$ determined by the uncertainty in the instrumental zero (1.5 s$^{-1}$) of the OH reactivity instrument. This uncertainty was applied in sensitivity runs of the model, but had a minor effect on the results discussed here.

Because of the unknown chemical nature of the background reactivity that dominates the loss of OH radicals for the zero air phase of the experiment, agreement between measured and modelled radical concentrations cannot be expected during this

initial phase. Therefore, no model calculation is shown for this part of the experiment.

Photolysis frequencies (j values) for O$_3$, NO$_2$, HONO, hydrogen peroxide (H$_2$O$_2$) and formaldehyde were constrained to the measurements. All the other photolysis frequencies present in the model were first calculated for clear sky conditions according to the MCM 3.3.1 and then scaled by the ratio of measured to calculated j(NO$_2$) to account for clouds and transmission of the chamber film. The model was constrained to measured water vapour, chamber pressure (= ambient

pressure), and temperature, NO, NO$_2$ and HONO. Values were re-initiated every minute. MBO and O$_3$ injections were implemented in the model by applying a source just active for the time of the injection. The O$_3$ source was adjusted to match the concentration measured at the injection and the MBO source to match the change in OH reactivity at the injection time. For completeness, the model included the reaction of MBO with O$_3$ although this reaction contributed on average 3% to the reactivity of MBO which was dominated by the reaction with OH radicals.

## 20  3 Results and discussion

### 3.1  Model comparison

Figure 2 shows the times series for the trace gases measured during the MBO experiment compared to the model including the sensitivity runs for the uncertainty introduced by the zero OH-reactivity value. At the beginning of the experiment, during the dark phase, formation of radicals was not expected as the roof was closed and only water vapour and ozone were

added. The reactivity of 1.7 s$^{-1}$ observed during this phase was due to desorption of trace gases from the walls of the chamber that could be observed during the humidification process. Some of these trace gases are HONO, HCHO and acetone as seen from their slow but steady increase. Immediately after opening of the roof, there was production of OH, HO$_2$ radicals and NO$_x$ from the photolysis of HONO and HCHO. After the injection of MBO, the OH reactivity was dominated by the reaction with MBO (~ 70% for all three MBO injections) and its oxidation products contributed significantly to the OH reactivity, up

to 40%, the more of the MBO reacted away. Good agreement between modelled and measured concentrations, well within the accuracy of the different instruments, could be observed for the majority of the species when MBO was oxidized by OH in this experiment. Formation of both measured major products from the oxidation of MBO, formaldehyde and acetone, was well reproduced by the model (averaged measurement to model ratio of 1.00 ± 0.02 for both). The modelled OH fitted the observation with an average measurement to model ratio of 1.0 ± 0.2 and the agreement between modelled and measured

HO$_2$ was, although less good compared to the OH, still satisfying (0.9 ± 0.1). The MBO decays due to its reaction with OH radicals were slightly over-predicted by the model (average observed to model ratio of 1.3 ± 0.2) in accordance with the measured decline of OH reactivity. This is in agreement with the PTR-TOF-MS measurement. However, results did not change significantly if the model was constrained to measured MBO concentrations.

The major uncertainties in this measurements-model comparison are introduced by the uncertainty of the zero measurement

of the OH reactivity data and by the possible interference of RO$_2$ radicals in the measured HO$_2$ signal. The first mostly affects the agreement between measured and modelled results for the OH reactivity itself. Modelled OH and HO$_2$ radicals are




also partially affected but the uncertainty introduced is lower than the accuracy of radical measurements while the remaining modelled species are not influenced. As mentioned in the instrumental description section, 20% is the upper limit for the interference from $RO_2$ radicals that could be expected from MBO on the $HO_2$ signal, based on the experiments performed with isoprene (Fuchs et al., 2011), for the conditions the instrument was run. The $HO_2$ concentration obtained from the model when accounting for this $RO_2$ interference would be, on average, only 8% larger ($\sim 5.5 \times 10^7$ $cm^{-3}$) than the $HO_2$ concentration without any $RO_2$ interference for the periods in the experiment where the MBO was present in the chamber. This value is lower than the accuracy of the $HO_2$ measurement itself and has an insignificant impact on the other trace gases.

### 3.2 Model comparison including hydrogen shift reactions

In a recent theoretical work from Knap et al. (2016), hydrogen shift reactions (H-shift) in the peroxy radicals originated after photooxidation of four different methyl-buten-ol isomers were investigated. The 1,4, 1,5 and 1,6 H-shift reactions were studied and the rate coefficients at ambient temperature and pressure were given. For the photooxidation of the MBO isomer under investigation in this study, predicted products are OH and $HO_2$ radicals, 2-hydroxy-2-methylpropanal (HMPR), acetone, and glycolaldehyde (Fig. 1). Also $\beta$- and $\delta$-epoxides are proposed as possible products although, due to their extremely slow unimolecular rate coefficients (Fig. 1), they are insignificant. As a sensitivity study, the three H-shift reactions, excluding the branching towards the epoxides, were included in the MCM 3.3.1 model as shown in Fig. 1, using the upper limit rate coefficients at 1013 hPa and 298 K as calculated by Knap et al. (2016). In the model, the H-shift reactions proceed directly to the final stable products with no formation of any intermediate. As expected from the low reaction rates for these reactions, their addition to the MBO degradation scheme has a very small impact with a change of less than 1% on any of the trace gases modelled in our chamber study bringing no improvement in the already good agreement between measurements and model calculations. This is consistent with the study by Knap et al. (2016) where they concluded that H-shift reactions are not relevant for the oxidation scheme of MBO even for low NO conditions (< 50 pptv) where the reaction with $HO_2$ remains the dominant loss process for the MBO-$RO_2$ radicals. This is also expected as MBO contains only one double bond and the fast H-shift reactions observed for isoprene and methacrolein are favoured by the formation of conjugated double bonds in the stable radical co-products (Peeters and Nguyen, 2012).

### 3.3 OH and $HO_2$ radicals budget analysis

The calculation of the experimental OH budget helps identifying possible missing OH sources, trusting on the correctness of the measured OH concentration and OH reactivity. The total experimental OH loss rate, $L_{OH}$, is given by the product of the OH concentration and the OH reactivity and, as the OH radical is assumed to be in steady-state, it should be equal to the total OH production rate ($P_{OH}$) (Eq. 1). $P_{OH}$ includes the OH production rate from known sources, $P_{OH}$Meas (Eq. 2), plus other possible sources. $L_{OH}$ can be compared with $P_{OH}$Meas, which can be calculated from the measured data.

$$L_{OH} = k_{OH} \times [OH] \approx P_{OH} = P_{OH}Meas + other\ sources \qquad \text{Eq. 1}$$

$$P_{OH}Meas = \left([HO_2] \times [NO] \times k_{HO_2+NO}\right) + \left([HONO] \times j(HONO)\right) + \left([O_3] \times j(O^1D) \times y\right) + \left([HO_2] \times [O_3] \times k_{HO_2+O_3}\right)$$

Eq. 2

Here [OH], [$HO_2$], [NO], [HONO], and [$O_3$] represent the measured concentrations of the trace gases, $k_{HO_2+NO}$ and $k_{HO_2+O_3}$ the rate coefficient of $HO_2$ with NO and ozone, respectively, j(HONO) and j(O1D) the photolysis rates of HONO and $O_3$, respectively, and y is the fraction of $O(^1D)$ reacting with water vapour multiplied with the OH yield of the $O(^1D)$ + $H_2O$ reaction. If all the sources contributing to the OH production are included in the calculation, then $P_{OH}$Meas $\approx P_{OH}$. In this study, the known OH sources considered are: reaction of NO and $HO_2$ ; reaction of $O_3$ and $HO_2$; photolysis of HONO;





photolysis of $O_3$, where y is the fraction of $O(^1D)$ reacting with water vapour multiplied with the OH yield of the $O(^1D)$ + $H_2O$ reaction. The formation of OH from ozonolysis of MBO is not included as it does not contribute noticeably.

Figure 3 shows the comparison between $P_{OH}Meas$ and the total experimental OH loss, $L_{OH}$. The averaged ratio between $P_{OH}Meas$ and $L_{OH}$ is $0.9 \pm 0.1$ (1σ). A small deviation from unity, which would indicate a missing OH source contributing at

most 20% to the total OH production, is obtained. Nevertheless, if the errors of the different measurements are taken into account, this deviation becomes not significant. For example, the total error of the total experimental OH loss is ~ 25% to which the errors of the measured traces gases, mainly of the $HO_2$ radicals (16%) and of the rate coefficients (~ 10%) used to calculate the $P_{OH}Meas$, should be added. From these considerations, the experimental OH budget can be considered closed and no additional OH sources aside the ones considered in Equation 2 is needed to explain the OH radicals loss.

Figure 3 also depicts the total modelled OH production $P_{OH}Mod$. This is included in the analysis of the experimental OH radical budget to understand how well the OH-formation paths in the model can describe the measurements. The averaged ratio between $P_{OH}Mod$ and $L_{OH}$ provides a value of $1.0 \pm 0.1$ (1σ). The good agreement observed between $P_{OH}Mod$ and $L_{OH}$ indicates that the model is able to correctly represent the OH radical sources. The averaged difference between $P_{OH}Mod$ and $P_{OH}Meas$ is $(2.3 \pm 1.9) \times 10^6$ cm$^{-3}$ s$^{-1}$. A large part of the difference, ~ $1.5 \times 10^6$ cm$^{-3}$ s$^{-1}$, is due to additional OH radical

sources included in the model and not considered in the experimental OH production, e.g. $RO_2$ ($CH_3CO_3$ and $HOCH_2CO_3$) reacting with $HO_2$ forming OH. The additional small discrepancy (~ $0.8 \times 10^6$ cm$^{-3}$ s$^{-1}$) is due to the differences observed for $HO_2$ and ozone between measurements and model calculations.

The analysis of the $HO_2$ budget is shown in Fig. 4. Here, differently from the OH budget, the measured $HO_2$ loss rate, $L_{HO_2}$, is compared to the total modelled $HO_2$ production rate, $P_{HO_2}Mod$. This comparison provides information on the

completeness in the understanding of the $HO_2$ production and loss processes for the MBO photooxidation mechanism. Within the model, the fifteen most important $HO_2$ production paths are explicitly considered and depicted in Fig. 4. The largest contribution to the $HO_2$ production comes from the decomposition of alkoxy radicals (46%), followed by the conversion of OH by the unknown background reactivity Y in the chamber (30%, not atmospherically relevant). Smaller contributions originate from formaldehyde photolysis (15%), and H-abstraction reaction by oxygen from the methoxy radical

($CH_3O$, 7%). As most of the relevant species contributing to the $HO_2$ production rate such as the alkoxy radicals were not measured and the background reactivity Y cannot be specified, it is not possible to calculate the production rate of $HO_2$ only from measured species as it was done for the OH radical budget.

The $HO_2$ radical is expected to be lost mainly via its reaction with NO accounting for ~ 90% of the total loss rate calculated from the model. Additional losses are $HO_2+HO_2$ self-reaction, reaction with ozone and reaction with the first generation $RO_2$

produced from the MBO oxidation (MBOAO2 and MBOBO2, Fig. 1). Therefore, the majority of the $HO_2$ loss rate can be obtained from measured NO, $HO_2$ and ozone concentrations (Eq. 3).

$$L_{HO_2}Meas = ([HO_2] \times [NO] \times k_{HO_2+NO}) + ([HO_2] \times [HO_2] \times k_{HO_2+HO_2}) + ([HO_2] \times [O_3] \times k_{HO_2+O_3}) \qquad \text{Eq. 3}$$

Here $[HO_2]$, $[NO]$, and $[O_3]$ represent the measured concentrations of the trace gases, $k_{HO_2+NO}$, $k_{HO_2+HO_2}$ and $k_{HO_2+O_3}$ the rate coefficient of $HO_2$ with NO, itself and $O_3$, respectively. The measured $HO_2$ loss rate, $L_{HO_2}Meas$, is in good agreement

with the total modelled $HO_2$ production rate with an average ratio of measured to modelled rates of $0.8 \pm 0.1$. The agreement with the total modelled $HO_2$ production increases (average ratio of measured to modelled rates of $0.9 \pm 0.1$) when including in the $L_{HO_2}Meas$ the modelled loss rate by the reaction of $HO_2$ with modelled $RO_2$ radicals. The largest discrepancies are observed during the first injection of MBO, because the calculated $HO_2$ production rate is smaller than what is obtained from the model. The main reason is the lower measured $HO_2$ concentration (13%) compared to the model during this period. With

the increasing agreement between modelled and measured $HO_2$ radical also the agreement in the $HO_2$ budget increases.

The good agreement observed in the $HO_2$ budget, although partly relying on modelled species concentrations, indicates that the $HO_2$ production, within this chamber experiment, can be explained by alkoxy radical decomposition, photolysis of formaldehyde, and the chamber specific Y source.



### 3.4 Comparison with previous studies

MBO was the major BVOC measured in two field campaigns which included measurements of OH and $HO_2$ radicals and a comparison with model calculations. The Biosphere Effects on Aerosols and Photochemisty Experiment II (BEARPEX09) campaign was performed near the Blodgett Forest Research Station in the California Sierra Nevada Mountains (Mao et al., 2012). This campaign was characterized by large MBO concentrations (daily average ~ 3000 pptv), followed by isoprene (daily average ~ 1700 pptv) and monoterpenes (α-pinene, daily average 100 pptv and β-pinene, daily average 70 pptv). Both measured OH and $HO_2$ radicals compared reasonably well with modelled calculations, in agreement with the results observed in this chamber study.

The 2010 Bio-hydro-atmosphere interactions of Energy, Aerosols, Carbon, $H_2O$, Organic and Nitrogen – Rocky Mountain Organic Carbon study (BEACHON-ROCS) was performed in the Manitou Experimental Forest in the Front range of the Colorado Rocky Mountains (Ortega et al., 2014). Here the dominant measured BVOC was MBO (daily average ~ 1600 pptv) followed by monoterpenes (daily average ~ 500 pptv) (Kim et al., 2013). As observed for the OH radical budget within this study, during the BEACHON-ROCS campaign the calculated OH concentration from ozone photolysis and from the recycling via $HO_2$ plus NO reaction, divided by the measured OH reactivity, agreed with the measured OH concentration (Kim et al., 2013). No additional OH recycling paths were necessary to close the OH budget. Nevertheless, during the BEACHON-ROCS campaign, the model was able to reproduce the OH radical concentration only when constrained to the $HO_2$ radical measurements as the model underestimated the measured $HO_2$ radicals up to a factor of 3 (Kim et al., 2013; Wolfe et al., 2014). This is different from what observed in the chamber experiment discussed in this study where a good agreement can be found between modelled and measured $HO_2$ concentration.

One difference between the two field studies is the BVOC compositions. During the BEARPEX09 campaign the concentration of the measured monoterpenes relative to the concentration of MBO during daytime was smaller (6%) compared to what observed during the BEACHON-ROCS campaign (31%). Two recent studies in environment with large concentrations of monoterpenes (Hens et al., 2014; Kaminski et al., 2017) also showed model calculations largely underestimating $HO_2$ radical measurement. Both studies concluded that the unaccounted $HO_2$ source seems to originate from monoterpene-oxidation products. The results collected by Hens et al. (2014) and by Kaminski et al. (2017) together with what is observed in this chamber study support that model-measurement discrepancies for $HO_2$ radicals in the BEACHON-ROCS campaign are not related to the MBO and its oxidation products but rather to the presence of monoterpenes and, as they were present in smaller concentrations, they would need to constitute a very efficient source of $HO_2$ radicals.

### 4 Summary and conclusions

A photooxidation experiment on 2-methyl-3-butene-2-ol (MBO), an important BVOC emitted by lodgepole and ponderosa pines, was performed in the atmospheric simulation chamber SAPHIR. Measurements of OH and $HO_2$ radicals and OH reactivity together with other important trace gases were compared to results from a state of the art chemical mechanistic model (MCM v3.3.1). The overall agreement is very good: firstly, an average observed to model ratio of 1.0 ± 0.2 and 0.9 ±0.1 is found for OH and $HO_2$ radicals, respectively. Also the MBO decay caused by reaction with OH radicals fits the expected decay from the model (average observed to model ratio of MBO concentration 1.3 ± 0.2) and is consistent with the measured OH reactivity. Moreover, the major measured products, acetone and formaldehyde, both match the model calculation with an average ratio of 1.00 ± 0.02. Addition of H-shift reactions from $RO_2$ radicals to the kinetic model as suggested in the literature (Knap et al., 2016) does not have a significant impact on the model results as expected from the small reaction rates (< 1.1 x $10^{-3}$ $s^{-1}$). The observed closure for both OH and $HO_2$ radical budgets indicates that their chemistry is well described by our current understanding of the MBO OH-initiated degradation processes.



The good agreement within the experimental OH budget is consistent with what was observed in previous field campaigns where MBO was the dominant BVOC measured (Mao et al., 2012; Kim et al., 2013). However there was no closure for the $HO_2$ budget or agreement between measurements and model results when a larger concentration of monoterpenes was also observed (Wolfe et al., 2014). This discrepancy cannot be explained by MBO photo-oxidation as a good agreement between

5 measured and calculated concentration of $HO_2$ is found in this chamber study. As a large discrepancy was also observed for chamber studies with $\beta$-pinene (Kaminski et al., 2017) and in environments with large monoterpenes concentrations (Hens et al., 2014), it is reasonable to assume that field observation for $HO_2$ radicals could be explained by an additional $HO_2$ radical source from monoterpenes-oxidation products, as proposed by Kaminski et al. (2017).

**Acknowledgments**

10 This work was supported by the EU FP-7 program EUROCHAMP-2 (grant agreement no. 228335). This project has received funding from the European Research Council (ERC) under the European Union's Horizon 2020 research and innovation program (SARLEP grant agreement no. 681529).



Table 1 Instrumentation for radical and trace-gas quantification during the MBO oxidation experiment.

| | Technique | Time resolution | 1σ precision | 1σ accuracy |
|---|---|---|---|---|
| **OH** | LIF | 47 s | $0.3 \times 10^6$ cm$^{-3}$ | 13% |
| **OH** | DOAS | 200 s | $0.8 \times 10^6$ cm$^{-3}$ | 6.5% |
| **HO$_2$** | LIF | 47 s | $1.5 \times 10^7$ cm$^{-3}$ | 16% |
| **$k_{OH}$** | Laser-photolysis + LIF | 180 s | 0.3 s$^{-1}$ | 10% |
| **NO** | Chemiluminescence | 180 s | 4 pptv | 5% |
| **NO$_2$** | Chemiluminescence | 180 s | 2 pptv | 5% |
| **HONO** | LOPAP | 300 s | 1.3 pptv | 12% |
| **O$_3$** | UV-absorption | 10 s | 1 ppbv | 5% |
| **VOCs** | PTR-TOF-MS | 30 | > 15 pptv | < 14% |
| **MBO** | GC | 30 min | 4 – 8 % | 5% |
| **HCHO** | Hantzsch monitor | 120 s | 20 pptv | 5% |



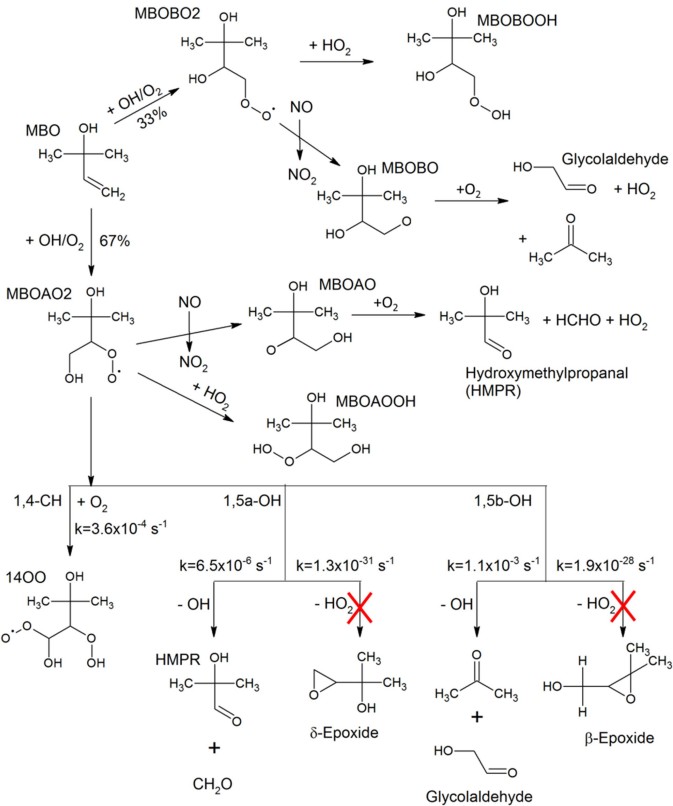

**Figure 1: Simplified MBO OH-oxidation reaction scheme as described in the MCM version 3.3.1, including 1,4 and 1,5 H-shift reactions and their rate coefficient at 1013 hPa and 298 K as suggested by Knap et al. (2016). These H-shift reactions were added to the MCM version 3.3.1 kinetic model for a sensitivity test excluding the ones forming the epoxides (marked with the red cross).**



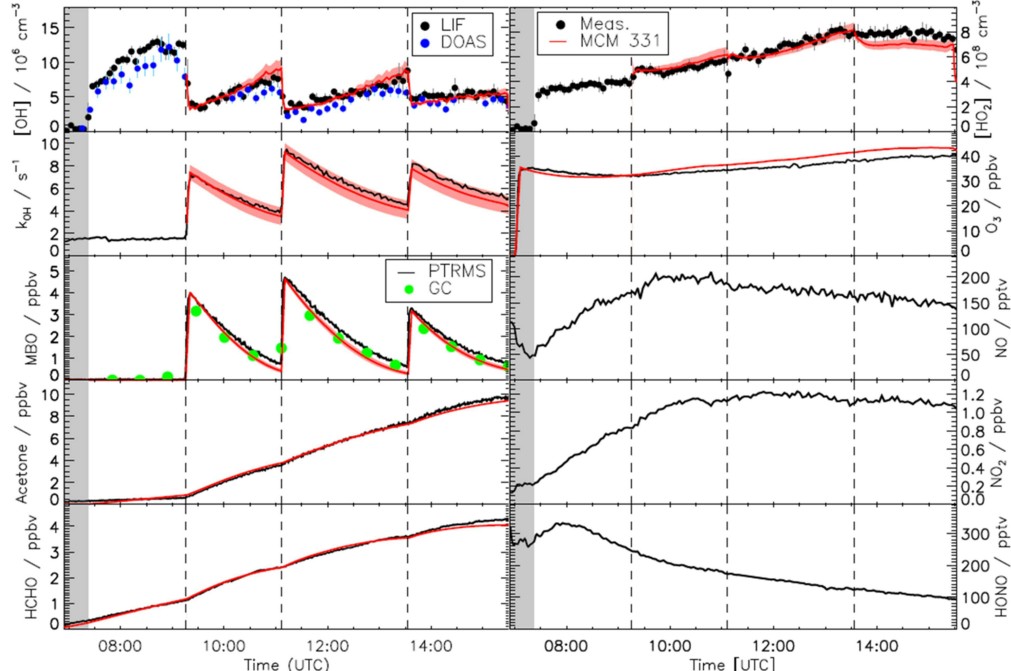

**Figure 2: Measured time series of OH, HO₂, MBO, acetone, formaldehyde, O₃, trace gases and OH reactivity compared to results obtained from modelling using the MCM version 3.3.1. There are no model results for NO, NO₂, and HONO as the model was constrained to the measurements. The red shaded areas represent the uncertainty of the model due to the uncertainties on the zero of the OH reactivity measurements (see text for details). Grey shaded areas indicate the times before opening the chamber roof and vertical dashed lines indicate the times when MBO was injected.**



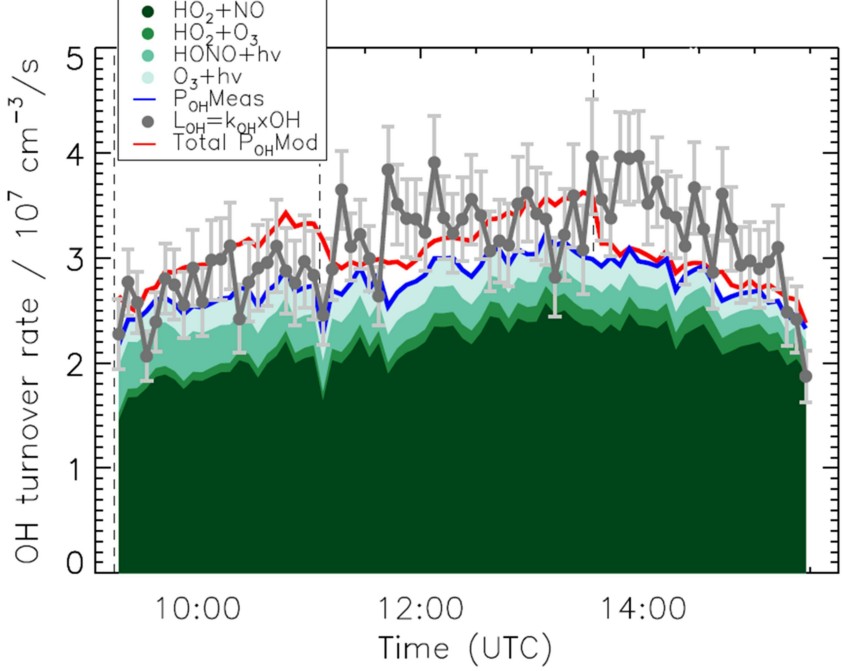

**Figure 3: OH budget for the MBO experiment. The experimentally determined total OH loss rate, $L_{OH}$, and individual production terms are shown. For comparison, the red line indicates the total modelled OH production, $P_{OH}Mod$, which equals the modelled loss rate. Vertical dashed lines indicate the times when MBO was injected. Error bars (1σ) for $L_{OH}$ include the accuracy of measurements.**





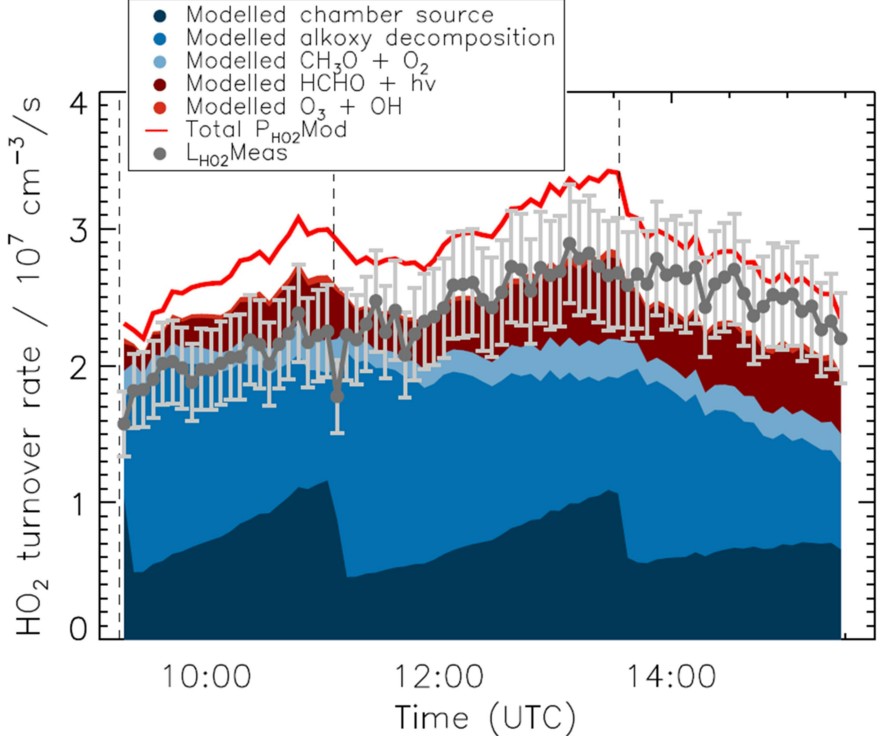

**Figure 4: HO$_2$ budget for the MBO experiment. The loss rate of HO$_2$ calculated from measured NO, ozone and HO$_2$ concentrations,             , and individual modelled production terms are shown. For comparison, the red line indicates the total modelled HO$_2$ production,             , which equals the total modelled loss rate. Vertical dashed lines indicate the times when MBO was injected. Error bars (1σ) for             include the accuracy of HO$_2$ measurements.**

20





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
