# Peer review of "Evaluation of OH and HO2 concentrations and their budgets during photo-oxidation of 2-methyl-3-butene-2-ol (MBO) in the atmospheric simulation chamber SAPHIR"

_Atmospheric Chemistry and Physics, 2018_

## Referee Comment (RC1) · Anonymous Referee #1 · 17 Apr 2018

This paper presents measurements of OH, HO2 and OH reactivity during the oxidation of 2-methyl-3-butene-2-ol (MBO) in the SAPHIR chamber. Measurements of OH and HO2 using a Laser-Induced Fluorescence - Fluorescence Assay by Gas Expansion (LIF-FAGE) instrument. While potential interferences with the OH measurements were not measured, the LIF-FAGE measurements were in good agreement with OH measurements by differential optical absorption spectroscopy (DOAS) in the SAPHIR chamber, suggesting that interferences associated with the FAGE technique were minimal. Measurements of HO2 were done at relatively low HO2+NO conversion efficien-

cies to minimize potential interferences from RO2 radicals.

The authors find that modeled concentrations of OH and HO2 concentrations during the oxidation of MBO using the latest version of the Master Chemical Mechanism were in good agreement with the measurements. The modeled total OH reactivity was also consistent with measurements of total OH reactivity as well as with measured total OH production, suggesting that the sources and sinks of OH were known and accounted for during the experiments. The modeled HO2 production rate was also found to be in good agreement with the measured HO2 loss rate, suggesting that the sources and sinks of this radical were accounted for during the oxidation experiments.

The results of these experiments are consistent with ambient measurements of OH and HO2 in forests where MBO was the dominant biogenic emission, where modeled radical concentrations were found to be in good agreement with the measurements. The results presented here suggest that discrepancies between measurements and models in areas where MBO is not the dominate emission are not due to uncertainties associated with the MBO oxidation mechanism. Rather, the discrepancies may be due to uncertainties in the oxidation of other biogenic emissions such as the oxidation of monoterpenes leading to HO2 production.

The paper is well written and the results suitable for publication in ACP after the authors have addressed the following:

1) The authors state that the largest uncertainty associated with the measurement – model agreement is the uncertainty associated with measurements of the OH reactivity during zero-air measurements, but it does not affect the modeled radical concentrations beyond the calibration accuracy of the measurements (pages 5-6). Unfortunately, Figure 2 does not display the model predictions of OH or HO2 during the zero measurements after the chamber has been opened. The authors state that (page 5) "Because of the unknown chemical nature of the background reactivity that dominates the loss of OH radicals for the zero air phase of the experiment, agreement between

measured and modelled radical concentrations cannot be expected during this initial phase. Therefore, no model calculation is shown for this part of the experiment." However, this unknown background reactivity has been observed previously as referenced in the manuscript (Fuchs et al., 2014; Kaminski et al., 2017) and in these studies the authors were able to demonstrate reasonable agreement of the model with the measurements during the zero-air experiments prior to addition of the reactant of interest. It is unclear why the authors were unable to reproduce the initial OH and HO2 concentrations in these experiments. Because the OH and HO2 concentrations depend on radical production from the background production of HONO and HCHO as well as O3 photolysis, illustrating the ability of the model to reproduce the chamber's OH and HO2 production during the zero-air experiments would give more confidence in the ability of the model to reproduce the radical concentrations after addition of MBO. This should be clarified in the revised manuscript.

2) Similarly, a significant fraction of the modelled HO2 radical budget appears to be due to the chamber source, which depends on the concentration of OH in the chamber. The authors state that the source is not "atmospherically relevant" (page 7). While I understand that this source is not relevant in the real atmosphere, do the authors mean that it is not a significant source in these chamber experiments, as it accounts for only 30% of the HO2 production? This could be clarified. What would the budget (and the modeled HO2 concentrations) look like if this source is removed?

3) Related to this, the authors assume that the chamber source acts similarly to the OH + CO reaction and propagates OH directly to HO2. Have the authors done a sensitivity test where they assume the unknown OH reactivity leads to either RO2 production or to OH radical termination instead of propagation directly to HO2? While the resulting modeled HO2 concentrations appear relatively insensitive to the uncertainty associated with the OH reactivity measurement, it is not clear whether the modeled HO2 concentrations and budget are sensitive to the assumption of the nature of the missing OH reactivity in the chamber. A missing reactivity mechanism that does not directly convert OH to HO2 may lead to a model underestimation of the measured HO2 concentrations, suggesting a missing HO2 source similar to that found in the ambient measurements discussed in section 3.4. Again, demonstrating that the model can reproduce the observed OH and HO2 during the zero-air experiments would provide some justification for their assumption of the nature of the missing OH reactivity in the chamber, but a sensitivity study that shows that the modeled HO2 concentrations were relatively insensitive to the nature of the missing OH reactivity would also give confidence in their conclusion that the chemistry of OH and HO2 is "well described our current understanding of the MBO OH-initiated degradation processes."

---

## Referee Comment (RC2) · Anonymous Referee #2 · 12 May 2018

Review of "Evaluation of OH and $HO_2$ concentrations and their budgets during photo-oxidation of 2-methyl-3-butene-2-ol (MBO) in the atmospheric simulation chamber SAPHIR" by Novelli et al.

This paper describes the data and analysis from a chamber experiment in which MBO was oxidized in the presence of $NO_x$. Measurements of $HO_x$ radicals, oxides of nitrogen, ozone and organic compounds were made as functions of time during the experiment that included three additions of MBO spaced about two hours apart. The observations were compared with calculations using the Master Chemical Mechanism model modified to include hydrogen shift reactions of peroxy radicals. From the comparisons, it is concluded that oxidation of MBO is well-represented by MCM, and that observed levels of OH and $HO_2$ in two studies that exceeded calculated amounts cannot be explained by radical recycling in the oxidation of MBO.

General comments.

While the agreement between observations and the MCM model is impressive, this reviewer is concerned that a single experiment may not cover a sufficient range of conditions (e.g. NO, $O_3$ and MBO levels, j-values) to fully test the MCM MBO oxidation mechanism. Because of this, it is premature to conclude that MBO oxidation chemistry is fully understood. It is important to conduct experiments at very low NO levels, so that the other peroxy radical chemistry can compete with oxidation by NO. The background chamber chemistry, while having been characterized and discussed in the literature, is also of concern. Since the nature of the OH reactivity is not fully understood, one is left to wonder whether the specie(s) contributing are also impacting the types and chemistry of peroxy radicals produced in the chamber. A measurement of $RO_2$ would help clarify the radical chemistry. Also, presentation of attempts to model the background chemistry (that were not shown because of "the unknown chemical nature of the background reactivity") would give the reader an indication of its cause and behavior, obviously without repeating everything in previous papers on this topic.

This reviewer is willing to accept the paper (with revisions) based just on the one experiment, but the conclusions would be much stronger with additional chamber runs. Suggest the authors consider more experiments before publication.

Specific comments.

Abstract.  In reading the first half of the abstract, this review was confused whether the discussion was related to the present study. Suggest adding something like "Several previous field studies" to indicate that this text is background information for the present study.

Page 1, line 17. Suggest "…studies have reported unexpectedly large…"

Page 1, line 18. Suggest "…mechanisms that largely…" or just eliminate "…which largely underestimated the observations", since unexpectedly large observations and model underestimations are essentially the same thing.

Page 1, line 22. Suggest "…radical concentrations showed…"

Page 1, line 28. Suggest "…trace gases agreed well…"

Page 1, line 30. Suggest "…cannot contribute in reconciling the unexplained…"

Page 2, line 17. Suggest "…radical concentrations measured…"

Page 2, line 40. Suggest "…within the Western US…"

Page 3, line 26. Suggest "…synthetic air until…"

Page 3, line 30. The statement about ozone duplicates what was stated in lines 27-28. Suggest consolidating.

Page 4, line 7. Suggest "…agreement between these two instruments…"

Page 4, lines 23-25. Is it possible to obtain concentrations of other hydrocarbon products that are expected (e.g. glycoaldehyde, HMPR, and others)? Also, do the PTR mass spectra offer any insight into

the missing OH reactivity? A statement as to why certain species are observed and others are not would be instructive for the reader.

Page 5. Line 9. This reviewer does not agree that agreement between measured and modeled concentrations cannot be expected during the initial phase (before MBO) addition. It is possible to constrain the nature of the OH reactant(s) by including a "CO-like" reactant and a "hydrocarbon-like" reactant. Suggest including the modeling results from the initial phase.

Page 5, line 30. Suggest "…more of the MBO that reacted…"

Page 5, line 35. Suggest "…although poorer than OH, was still satisfactory…"

Page 6, line 18. Suggest "…of any intermediates."

Page 6, line 28. Suggest "…missing OH sources, assuming the correctness…"

Page 7, line 1-2. Suggest removing "where y is the fraction of $O(^1D)$ reacting with water vapour multiplied with the OH yield of the $O(^1D) + H_2O$ reaction." This is given in page 6, lines 38-39.

Page 7, line 6. Suggest "…deviation becomes insignificant."

Page 7, line 18. Suggest "Here, different from the…"

Page 7, line 22. Suggest more description of what is meant by decomposition of alkoxy radicals, perhaps showing a sample reaction.

Page 7, line 26. Yes, the alkoxy radicals were not measured, but their abundance can be inferred from the rates of reaction of OH with hydrocarbons. This review suggests that a production of $HO_2$ can be calculated using steady-state assumptions for alkoxy radicals. Also, modeling exercises described earlier can help better constrain the background reactivity.

Page 7, line 36-37. Suggest "…when including the $L_{HO2, meas}$…"

Page 8, lines 1-20. When discussing BEARPEX and BEACHON-ROCS measure-model comparisons, it is important to identify whether the models used the same mechanisms as that used in this study. Suggest adding one or sentences to clarify this.

Page 8, line 32. Suggest "…state-of-the-art…"

Page 8, line 33-34. Suggest "…firstly, average observed to modeled ratios of 1.0 ± 0.2 and 0.9 ± 0.1 are found…"

Page 8, line 34-35. The observed to modeled MBO being near 1.0 does not mean the observed and modeled decays are in agreement. A fit of the ratios versus time for the observations and model would be better. Alternatively, a comparison of lifetimes for each of the decays could be instructive.

Page 8, line 38. Suggest "…do not have significant impacts on the model…"

Page 9, line 4. Suggest "…photo-oxidation as good agreement…"

Page 9, line 5. Suggest "As large discrepancies were also observed…"

Page 10. Suggest a note in the table to indicate that accuracy estimates are for concentrations well above the detection limit (which is three times the precision? Or ?). Also, suggest a note to indicate the PTR precision and accuracy values vary with compound. Could give a range of values rather than > and < values.

Page 15, References. Suggest using hanging paragraph indent, as it is difficult to find a specific reference with the current format.

---

## Author Comment (AC1) · 27 Jun 2018

**We thank the anonymous referees for reading the paper carefully and providing thoughtful comments, which have resulted in improvements in the revised version of the manuscript. We reply to each comment below in bold text.**

**Anonymous Referee #1**

This paper presents measurements of OH, HO2 and OH reactivity during the oxidation of 2-methyl-3-butene-2-ol (MBO) in the SAPHIR chamber. Measurements of OH and HO2 using a Laser-Induced Fluorescence - Fluorescence Assay by Gas Expansion (LIF-FAGE) instrument. While potential interferences with the OH measurements were not measured, the LIF-FAGE measurements were in good agreement with OH measurements by differential optical absorption spectroscopy (DOAS) in the SAPHIR chamber, suggesting that interferences associated with the FAGE technique were minimal. Measurements of HO2 were done at relatively low HO2+NO conversion efficiencies to minimize potential interferences from RO2 radicals.

The authors find that modeled concentrations of OH and HO2 concentrations during the oxidation of MBO using the latest version of the Master Chemical Mechanism were in good agreement with the measurements. The modeled total OH reactivity was also consistent with measurements of total OH reactivity as well as with measured total OH production, suggesting that the sources and sinks of OH were known and accounted for during the experiments. The modeled HO2 production rate was also found to be in good agreement with the measured HO2 loss rate, suggesting that the sources and sinks of this radical were accounted for during the oxidation experiments.

The results of these experiments are consistent with ambient measurements of OH and HO2 in forests where MBO was the dominant biogenic emission, where modeled radical concentrations were found to be in good agreement with the measurements. The results presented here suggest that discrepancies between measurements and models in areas where MBO is not the dominate emission are not due to uncertainties associated with the MBO oxidation mechanism. Rather, the discrepancies may be due to uncertainties in the oxidation of other biogenic emissions such as the oxidation of monoterpenes leading to HO2 production.

The paper is well written and the results suitable for publication in ACP after the authors have addressed the following:

1) The authors state that the largest uncertainty associated with the measurement – model agreement is the uncertainty associated with measurements of the OH reactivity during zero-air measurements, but it does not affect the modeled radical concentrations beyond the calibration accuracy of the measurements (pages 5-6). Unfortunately, Figure 2 does not display the model predictions of OH or HO2 during the zero measurements after the chamber has been opened. The authors state that (page 5) "Because of the unknown chemical nature of the background reactivity that dominates the loss of OH radicals for the zero air phase of the experiment, agreement between measured and modelled radical concentrations cannot be expected during this initial phase. Therefore, no model calculation is shown for this part of the experiment." However, this unknown background reactivity has been observed previously as referenced in the manuscript (Fuchs et al., 2014; Kaminski et al., 2017) and in these studies the authors were able to demonstrate reasonable agreement of the model with the measurements during the

zero-air experiments prior to addition of the reactant of interest. It is unclear why the authors were unable to reproduce the initial OH and HO2 concentrations in these experiments. Because the OH and HO2 concentrations depend on radical production from the background production of HONO and HCHO as well as O3 photolysis, illustrating the ability of the model to reproduce the chamber's OH and HO2 production during the zero-air experiments would give more confidence in the ability of the model to reproduce the radical concentrations after addition of MBO. This should be clarified in the revised manuscript.

**The authors would like to clarify the challenges in modelling the radical concentrations during the zero-air phase of experiments in the SAPHIR chamber. As the referee correctly underlines, there is production of radical species from HONO, HCHO and $O_3$ photolysis but not all of the OH radical reactants are known. After accounting for the known OH reactants, a large fraction of the measured OH reactivity during the zero-air phase (~ 70%) remains unexplained. To close the gap between the OH reactivity measurement and the model results in the zero-air phase, CO is chosen as the OH reactant with a direct formation of $HO_2$ radicals and its concentration is kept constant throughout the experiment. This is of course a simplified approach and its purpose is not to correctly reproduce the chemistry of the radical. Therefore a perfect agreement between radical measurements and model results is not expected. The authors would like to emphasize that after the injection of MBO the oxidation chemistry in the chamber is determined by MBO and its products as even when most of the MBO has reacted away, the unexplained background reactivity contributes to less than a third to the total OH reactivity. The zero-air phase mainly serves to check the status of the chamber to identify, for example, if there are any unexpected contaminations. This is now better clarified in the manuscript.**

[Figure]

Figure 1. Measured time series including the modelling results during the zero-air phase.

**Model calculations during the zero-air phase for this study have larger uncertainties compared to experiments shown in the past due to an increased uncertainty on the total OH reactivity measurement (± 0.6 s⁻¹ compared to ± 0.1 s⁻¹ observed in different experiments). As during the zero-air phase this uncertainty is a substantial fraction of the**

**total reactivity, it propagates into a large uncertainty on the OH radical concentration. This effect can be seen in figure 1. The shaded area represents the uncertainty observed when using 0.6 or 2.0 s$^{-1}$ as starting OH reactivity in the chamber. For the zero-air phase this results in an uncertainty of ~ 30 % which reduces to less than 5% once the MBO is injected and the total OH reactivity reaches 8 s$^{-1}$. The agreement observed in figure 1 between model results and radical measurements in the zero-air phase is comparable to what observed in previous studies (Fuchs et al., 2013; Fuchs et al., 2014; Kaminski et al., 2017). As after the injection of MBO the reactivity of OH and HO$_2$ radicals is dominated by MBO and its products, the conclusions of this study are not affected by the unknown reactivity during the zero-air phase. However, following the suggestion of both referees figure 4 (see answer to comment #3), which shows model results also for the initial zero-air phase, was added to the supplementary information of the manuscript.**

2) Similarly, a significant fraction of the modelled HO2 radical budget appears to be due to the chamber source, which depends on the concentration of OH in the chamber. The authors state that the source is not "atmospherically relevant" (page 7). While I understand that this source is not relevant in the real atmosphere, do the authors mean that it is not a significant source in these chamber experiments, as it accounts for only 30% of the HO2 production? This could be clarified. What would the budget (and the modeled HO2 concentrations) look like if this source is removed?

**The authors would like first to apologize to the referees as the wrong budget figure for the HO$_2$ radicals was shown in the manuscript. The budget for a reactivity of 2.0 s$^{-1}$ during the zero-air phase was shown instead of the average values as shown in figure 1. The figure was modified (Figure 2).**

[Figure]

Figure 2. HO$_2$ budget for the MBO experiment.

**With the comment "not atmospherically relevant" the authors wanted to underline that a certain fraction of HO$_2$ radicals, between 8 and 30%, on average, depending on the OH**

**reactivity, is produced by the reaction of OH with species Y and therefore is not related to the MBO chemistry but is specific to the SAPHIR chamber. The sentence was rephrased for clarity in the manuscript.**

**Following the suggestion of the referee, this source of HO$_2$ radicals was removed for a test-run of the model (Figure 3).**

[Figure]

Figure 3. HO$_2$ budget for the MBO experiment removing the Y source.

**When the chamber source Y is removed from the model, a larger contribution from the other sources to the HO$_2$ radical production can be observed and the agreement with the measurement is still good (Figure 3). A larger discrepancy would be observed in the agreement between OH and HO$_2$ radicals in the zero-air phase but after the injection of MBO the observed to model ratio for the HO$_2$ radicals would be on average 0.7 ± 0.2 which is still reasonable.**

**This is a confirmation that the chemistry observed during the zero-air phase of the experiment has a negligible impact once MBO is injected in the chamber.**

3) Related to this, the authors assume that the chamber source acts similarly to the OH + CO reaction and propagates OH directly to HO2. Have the authors done a sensitivity test where they assume the unknown OH reactivity leads to either RO2 production or to OH radical termination instead of propagation directly to HO2? While the resulting modeled HO2 concentrations appear relatively insensitive to the uncertainty associated with the OH reactivity measurement, it is not clear whether the modeled HO2 concentrations and budget are sensitive to the assumption of the nature of the missing OH reactivity in the chamber. A missing reactivity mechanism that does not directly convert OH to HO2 may lead to a model underestimation of the measured HO2 concentrations, suggesting a missing HO2 source similar to that found in the ambient measurements discussed in section 3.4. Again, demonstrating that the model can reproduce the observed OH and HO2 during the zero-air experiments would provide some justification for their assumption of the nature of the missing OH reactivity in the chamber, but a sensitivity study that shows that the modeled HO2 concentrations were relatively insensitive to the nature of the missing OH reactivity would also give confidence in their conclusion that the

chemistry of OH and HO2 is "well described our current understanding of the MBO OH-initiated degradation processes."

Following the suggestion of the referee, a test-run of the model using $CH_4$ as species Y instead of CO was performed. Figure 4 contains on the left panels a comparison for OH and $HO_2$ radicals, OH reactivity and MBO between model results and measurements when CO is used as species Y and on the right panels when using $CH_4$ as species Y instead. CO and $CH_4$ are used as they represent two extremes for the $HO_2$ formation after the reaction with OH. In the CO case the formation of $HO_2$ radicals is immediate, while with $CH_4$ there are several steps in between.

[Figure]

Figure 4. Measured time series including the modelling results during the zero-air phase using CO (left panels) or $CH_4$ (right panels) as Y species.

When $CH_4$ is used as species Y there is a slight improvement during the zero-air phase in the agreement between model results and measured OH radicals. An even larger improvement can be observed, during the zero-air phase, for the $HO_2$ radicals. Nevertheless, after the injection of MBO there is a negligible difference for all species. This shows that after the injection of MBO, the radical concentrations are pretty insensitive to the species Y confirming that the chemistry of OH and $HO_2$ radicals is well described by the current understanding of the MBO OH-initiated degradation processes. Figure 4 was added to the supplementary information of the manuscript.

Anonymous Referee #2

This paper describes the data and analysis from a chamber experiment in which MBO was oxidized in the presence of NOx. Measurements of HOx radicals, oxides of nitrogen, ozone and organic compounds were made as functions of time during the experiment that included three additions of MBO spaced about two hours apart. The observations were compared with calculations using the Master Chemical Mechanism model modified to include hydrogen shift reactions of peroxy radicals. From the comparisons, it is concluded that oxidation of MBO is well-represented by MCM, and that observed levels of OH and HO2 in two studies that exceeded calculated amounts cannot be explained by radical recycling in the oxidation of MBO.

General comments.
While the agreement between observations and the MCM model is impressive, this reviewer is concerned that a single experiment may not cover a sufficient range of conditions (e.g. NO, O3 and MBO levels, j-values) to fully test the MCM MBO oxidation mechanism. Because of this, it is premature to conclude that MBO oxidation chemistry is fully understood. It is important to conduct experiments at very low NO levels, so that the other peroxy radical chemistry can compete with oxidation by NO.

**A total of three experiments were performed at very similar conditions as the one analyzed in this work but two of them were affected by instrumental failures and are therefore not included in this study. However, the good agreement between modelled and measured radical and trace gases found in the experiment shown in this work and also the results from theoretical work, from which no effect from additional radical chemistry is expected (Knap et al., 2016), led us to the decision that there was no need to perform more MBO chamber experiments, which require a high effort compared to typical laboratory experiments. The purpose of this work was to investigate the chemistry of MBO under conditions expected in a rural environment. The values of ozone and NO used during this experiment are comparable with what was observed in the two MBO field campaigns compared with this study (~50 ppbv of ozone and 150 pptv of NO). Therefore it is reasonable to say that, at typical atmospheric conditions in which MBO was previously measured in forests, this work shows that the MBO oxidation chemistry is fully understood.**

The background chamber chemistry, while having been characterized and discussed in the literature, is also of concern. Since the nature of the OH reactivity is not fully understood, one is left to wonder whether the specie(s) contributing are also impacting the types and chemistry of peroxy radicals produced in the chamber. A measurement of RO2 would help clarify the radical chemistry. Also, presentation of attempts to model the background chemistry (that were not shown because of "the unknown chemical nature of the background reactivity") would give the reader an indication of its cause and behavior, obviously without repeating everything in previous papers on this topic.

**Please refer to the answers 1 and 3 provided to referee #1**

This reviewer is willing to accept the paper (with revisions) based just on the one experiment, but the conclusions would be much stronger with additional chamber runs. Suggest the authors consider more experiments before publication.

Specific comments.

Abstract. In reading the first half of the abstract, this review was confused whether the discussion was related to the present study. Suggest adding something like "Several previous field studies" to indicate that this text is background information for the present study.
**Changed as suggested.**

Page 1, line 17. Suggest "…studies have reported unexpectedly large…"
**Changed as suggested.**

Page 1, line 18. Suggest "…mechanisms that largely…" or just eliminate "…which largely underestimated the observations", since unexpectedly large observations and model underestimations are essentially the same thing.
**Changed as suggested.**

Page 1, line 22. Suggest "…radical concentrations showed…"
**Changed as suggested.**

Page 1, line 28. Suggest "…trace gases agreed well…"
**Changed as suggested.**

Page 1, line 30. Suggest "…cannot contribute in reconciling the unexplained…"
**Changed as suggested.**

Page 2, line 17. Suggest "…radical concentrations measured…"
**Changed as suggested.**

Page 2, line 40. Suggest "…within the Western US…"
**Changed as suggested.**

Page 3, line 26. Suggest "…synthetic air until…"
**Changed as suggested.**

Page 3, line 30. The statement about ozone duplicates what was stated in lines 27-28. Suggest consolidating.
**Changed as suggested.**

Page 4, line 7. Suggest "…agreement between these two instruments…"
**Changed as suggested.**

Page 4, lines 23-25. Is it possible to obtain concentrations of other hydrocarbon products that are expected (e.g. glycoaldehyde, HMPR, and others)? Also, do the PTR mass spectra offer any insight into the missing OH reactivity? A statement as to why certain species are observed and others are not would be instructive for the reader.
**The PTRMS was only calibrated for the species shown in the figures and as such concentrations of other products are not available.**

Page 5. Line 9. This reviewer does not agree that agreement between measured and modeled concentrations cannot be expected during the initial phase (before MBO) addition. It is possible to constrain the nature of the OH reactant(s) by including a "CO-like" reactant and a "hydrocarbon-like" reactant. Suggest including the modeling results from the initial phase.
**Please refer to the answers to the first and third point of the referee #1. Following the comments of both referees a figure including the model results during the zero-air phase**

**for both a CO and a hydrocarbon-like reactants was added to the supplementary information.**

Page 5, line 30. Suggest "…more of the MBO that reacted…"
**Changed as suggested.**

Page 5, line 35. Suggest "…although poorer than OH, was still satisfactory…"
**Changed as suggested.**

Page 6, line 18. Suggest "…of any intermediates."
**Changed as suggested.**

Page 6, line 28. Suggest "…missing OH sources, assuming the correctness…"
**Changed as suggested.**

Page 7, line 1-2. Suggest removing "where y is the fraction of O(1D) reacting with water vapour multiplied with the OH yield of the O(1D) + H2O reaction." This is given in page 6, lines 38-39.
**Changed as suggested.**

Page 7, line 6. Suggest "…deviation becomes insignificant."
**Changed as suggested.**

Page 7, line 18. Suggest "Here, different from the…"
**Changed as suggested.**

Page 7, line 22. Suggest more description of what is meant by decomposition of alkoxy radicals, perhaps showing a sample reaction.
**The sentence in the manuscript was changed as it was not accurate and a direct reference to the reaction schematic (Figure 1 of the manuscript) was added.**

Page 7, line 26. Yes, the alkoxy radicals were not measured, but their abundance can be inferred from the rates of reaction of OH with hydrocarbons. This review suggests that a production of HO2 can be calculated using steady-state assumptions for alkoxy radicals. Also, modeling exercises described earlier can help better constrain the background reactivity.
**The authors prefer to keep the budget for the HO$_2$ radicals as is since the calculation of a steady-state concentration of alkoxy radicals would have to rely heavily on the results from the model result and would not add additional insights.**

Page 7, line 36-37. Suggest "…when including the LHO2, meas…"
**$L_{HO_2}$Meas is used as one word to be easily identified in the legend of figure 3.**

Page 8, lines 1-20. When discussing BEARPEX and BEACHON-ROCS measure-model comparisons, it is important to identify whether the models used the same mechanisms as that used in this study. Suggest adding one or sentences to clarify this.
**A sentence to clarify was added.**

Page 8, line 32. Suggest "…state-of-the-art…"
**Changed as suggested.**

Page 8, line 33-34. Suggest "…firstly, average observed to modeled ratios of 1.0 ± 0.2 and 0.9 ± 0.1 are found…"
**Changed as suggested.**

Page 8, line 34-35. The observed to modeled MBO being near 1.0 does not mean the observed and modeled decays are in agreement. A fit of the ratios versus time for the observations and model would be better. Alternatively, a comparison of lifetimes for each of the decays could be instructive.
**As can be seen from the time series plot in Fig. 2, the modelled MBO decays a little bit faster than the MBO measured by the PTRMS. This ratio becomes the highest (~ 1.5) at the very end of the experiment. The authors do not think it is necessary to add an additional plot showing the time dependency of the ratio.**

Page 8, line 38. Suggest "…do not have significant impacts on the model…"
**Although we appreciate the suggestion of the referee, we prefer the current wording.**

Page 9, line 4. Suggest "…photo-oxidation as good agreement…"
**Changed as suggested.**

Page 9, line 5. Suggest "As large discrepancies were also observed…"
**Changed as suggested.**

Page 10. Suggest a note in the table to indicate that accuracy estimates are for concentrations well above the detection limit (which is three times the precision? Or ?). Also, suggest a note to indicate the PTR precision and accuracy values vary with compound. Could give a range of values rather than > and < values.
**For this experiment, accuracy and precision for the species measured by the PTRMS (MBO and acetone) were the same.**

Page 15, References. Suggest using hanging paragraph indent, as it is difficult to find a specific reference with the current format.
**Changed as suggested.**

**References**

Fuchs, H., Hofzumahaus, A., Rohrer, F., Bohn, B., Brauers, T., Dorn, H. P., Haseler, R., Holland, F., Kaminski, M., Li, X., Lu, K., Nehr, S., Tillmann, R., Wegener, R., and Wahner, A.: Experimental evidence for efficient hydroxyl radical regeneration in isoprene oxidation, Nat Geosci, 6, 1023-1026, doi:10.1038/Ngeo1964, 2013.

Fuchs, H., Acir, I. H., Bohn, B., Brauers, T., Dorn, H. P., Häseler, R., Hofzumahaus, A., Holland, F., Kaminski, M., Li, X., Lu, K., Lutz, A., Nehr, S., Rohrer, F., Tillmann, R., Wegener, R., and Wahner, A.: OH regeneration from methacrolein oxidation investigated in the atmosphere simulation chamber SAPHIR, Atmos. Chem. Phys., 14, 7895-7908, doi:10.5194/acp-14-7895-2014, 2014.

Kaminski, M., Fuchs, H., Acir, I. H., Bohn, B., Brauers, T., Dorn, H. P., Häseler, R., Hofzumahaus, A., Li, X., Lutz, A., Nehr, S., Rohrer, F., Tillmann, R., Vereecken, L., Wegener, R., and Wahner, A.: Investigation of the β-pinene photooxidation by OH in the atmosphere simulation chamber SAPHIR, Atmos. Chem. Phys., 17, 6631-6650, doi:10.5194/acp-17-6631-2017, 2017.

Knap, H. C., Schmidt, J. A., and Jorgensen, S.: Hydrogen shift reactions in four methyl-buten-ol (MBO) peroxy radicals and their impact on the atmosphere, Atmos. Environ., 147, 79-87, doi:10.1016/j.atmosenv.2016.09.064, 2016.

---

## Editor Decision (ED1)

**Referee 1.**

**Editor**. Happy with responses and the additional material added to the main MS and also the SI.

**Referee 2.**

Says:

**General comments.**
While the agreement between observations and the MCM model is impressive, this reviewer is concerned that a single experiment may not cover a sufficient range of conditions (e.g. NO, O3 and MBO levels, j‑values) to fully test the MCM MBO oxidation mechanism. Because of this, it is premature to conclude that MBO oxidation chemistry is fully understood. It is important to conduct experiments at very low NO levels, so that the other peroxy radical chemistry can compete with oxidation by NO.

**Your response is:**

A total of three experiments were performed at very similar conditions as the one analyzed in this work but two of them were affected by instrumental failures and are therefore not included in this study. However, the good agreement between modelled and measured radical and trace gases found in the experiment shown in this work and also the results from theoretical work, from which no effect from additional radical chemistry is expected (Knap et al., 2016), led us to the decision that there was no need to perform more MBO chamber experiments, which require a high effort compared to typical laboratory experiments. The purpose of this work was to investigate the chemistry of MBO under conditions expected in a rural environment. The values of ozone and NO used during this experiment are comparable with what was observed in the two MBO field campaigns compared with this study (~50 ppbv of ozone and 150 pptv of NO). Therefore it is reasonable to say that, at typical atmospheric conditions in which MBO was previously measured in forests, this work shows that the MBO oxidation chemistry is fully understood.

**Editor:** I like this response, but can you please incorporate some of this into the revised MS? I think it would be useful for readers to see the reasons to justify why the three experiments are sufficient for this study.

**The referees also says:**
Page 4, lines 23‑25. Is it possible to obtain concentrations of other hydrocarbon products that are expected (e.g. glycoaldehyde, HMPR, and others)? Also, do the PTR mass spectra offer any insight into the missing OH reactivity? A statement as to why certain species are observed and others are not would be instructive for the reader.

**Your response:**
The PTRMS was only calibrated for the species shown in the figures and as such concentrations of other products are not available.

**Editor:**
Again I like the response, but can you incorporate some of this response into the revised MS so the reader can gain insight into this.

---

## Author Response (AR2)

**We thank the editor for reading the paper carefully and providing comments, which have resulted in improvements in the revised version of the manuscript. We reply to each comment below in bold text.**

**Referee 1.**

**Editor**. Happy with responses and the additional material added to the main MS and also the SI.

**Referee 2.**

Says:

**General comments.**

While the agreement between observations and the MCM model is impressive, this reviewer is concerned that a single experiment may not cover a sufficient range of conditions (e.g. NO, O3 and MBO levels, j-values) to fully test the MCM MBO oxidation mechanism. Because of this, it is premature to conclude that MBO oxidation chemistry is fully understood. It is important to conduct experiments at very low NO levels, so that the other peroxy radical chemistry can compete with oxidation by NO.

**Your response is:**

A total of three experiments were performed at very similar conditions as the one analyzed in this work but two of them were affected by instrumental failures and are therefore not included in this study. However, the good agreement between modelled and measured radical and trace gases found in the experiment shown in this work and also the results from theoretical work, from which no effect from additional radical chemistry is expected (Knap et al., 2016), led us to the decision that there was no need to perform more MBO chamber experiments, which require a high effort compared to typical laboratory experiments. The purpose of this work was to investigate the chemistry of MBO under conditions expected in a rural environment. The values of ozone and NO used during this experiment are comparable with what was observed in the two MBO field campaigns compared with this study (~50 ppbv of ozone and 150 pptv of NO). Therefore it is reasonable to say that, at typical atmospheric conditions in which MBO was previously measured in forests, this work shows that the MBO

oxidation chemistry is fully understood.

**Editor:** I like this response, but can you please incorporate some of this into the revised MS? I think it would be useful for readers to see the reasons to justify why the three experiments are sufficient for this study.

**After the editor suggestion, the following text was added on page 3 line 33 of the revised manuscript.**

**"Two additional experiments where performed at very similar conditions but due to instrumental failures they could not be included in this study. The experiment shown is composed of three independent injections of MBO and the range of NO and $O_3$ in the chamber is analogous to what observed in the field studies where large concentrations of MBO were measured (~50 ppb of $O_3$ and 150 pptv of NO) giving confidence that what observed in this study can be**

**compared to ambient data."**

**The referees also says:**

Page 4, lines 23-25. Is it possible to obtain concentrations of other hydrocarbon products that are expected (e.g. glycoaldehyde, HMPR, and others)? Also, do the PTR mass spectra offer any insight into the missing OH reactivity? A

statement as to why certain species are observed and others are not would be instructive for the reader.

**Your response:**

The PTRMS was only calibrated for the species shown in the figures and as such concentrations of other products are not available.

**Editor:**

Again I like the response, but can you incorporate some of this response into the revised MS so the reader can gain insight into this.

**The following text was added on page 4 line 31 of the revised manuscript.**

[revised manuscript text omitted]